**Data Availability Statement:** All relevant data are within the manuscript and its Supporting Information files.

**Funding:** The author(s) received no specific funding for this work.

# Pharmacist-led medication reviews: A scoping review of systematic reviews

**Miriam Craske**[ID][1]*, **Wendy Hardeman**[2], **Nicholas Steel**[3], **Michael James Twigg**[1¤]

**1** School of Pharmacy, University of East Anglia, Norwich, England, **2** School of Health Sciences, University of East Anglia, Norwich, England, **3** Norwich Medical School, University of East Anglia, Norwich, England

¤ Current address: Research and Evaluation Team, NHS Norfolk and Waveney, Norwich, England
* m.craske@uea.ac.uk

## Abstract

### Background

Medication reviews aim to support patients who take medicines, and they are often led by pharmacists. There are different types of medication reviews undertaken in various settings. Previous research undertaken in 2015 found mixed evidence that medication reviews in community settings improve clinical outcomes, but further work needs to be undertaken to establish their impact on patient-orientated and economic outcomes.

### Aim

This scoping review aims to explore the extent and range of systematic reviews of medication reviews conducted by pharmacists, the nature of the intervention, the evidence for effectiveness, and reported research gaps.

### Method

Systematic reviews were included irrespective of participants, settings or outcomes and were excluded if pharmacists did not lead the delivery of the included interventions. Data extracted included the design of included studies, population, setting, main results, description of interventions, and future research recommendations.

### Results

We identified twenty-four systematic reviews that reported that medication review interventions were diverse, and their nature was often poorly described. Two high-quality reviews reported that there was evidence of no effect on mortality; of these one reported an improvement in medicines-related problems (all studies reported an increase of identified problems), and another a reduction in hospital readmissions (Risk ratio 0.93 [95% CI 0.89, 0.98]). Other lower-quality reviews reported evidence supporting intervention effectiveness for some clinical outcomes (odds ratio: achieving diabetes control = 3.11 95% prediction intervals (PI), 1.48–6.52, achieving blood pressure target = 2.73, 95% PI, 1.05–7.083.50).

### Conclusion

There is mixed evidence of effectiveness for medication reviews across settings and patient populations. There is limited data about the implementation of medication reviews, therefore

**Competing interests:** The authors have declared that no competing interests exist.

is difficult to ascertain which components of the intervention lead to improved outcomes. As medication reviews are widely implemented in practice, further research should explore the nature of the interventions, linking the components of these to outcomes.

## Introduction

Medication reviews (MRs) are a recognised intervention undertaken by healthcare professionals, including pharmacists, to support patients prescribed multiple medicines or with complex medication regimes [1]. The concept of different types of medication review was first introduced in 2002 [2]; this was expanded upon by the National Prescribing Centre in 2008 [3]. Medication reviews are classified as a prescription, concordance and compliance, or clinical review [3]. Medication reviews are often led by pharmacists in different practice settings. An overview review by Jokanovic et al. [4] synthesised the evidence for outcomes of pharmacist-led medication reviews in community settings.

Given that additional systematic reviews have been published since the Jokanovic review in 2017 [4], it is prudent to undertake a scoping review to describe the most recent evidence of pharmacist-led medication reviews in all settings and populations. Scoping reviews can be used to determine the coverage of a body of literature and help identify potential questions for future systematic reviews [5]. A scoping review is justified to identify the extent (geographical location, medication review recipients in primary studies), range (study design, type of intervention included in the systematic reviews), and nature (description, attributes/ components of MR) of research in this field to provide an overview of the latest evidence and inform future research. Given the number of systematic reviews already published in this field, this scoping review will focus on these reviews and not the primary research, to avoid repetition of existing work.

Therefore, our research question is, what is the systematic review evidence about the nature and effectiveness of medication review interventions conducted by pharmacists, and what are the gaps in research knowledge?

### Aim

To describe evidence from existing systematic reviews on pharmacist-led medication reviews to inform future research. This will be achieved by addressing the following objectives:

- to describe the extent and range of pharmacist-led medication reviews

- report MR nature as described in the literature

- describe evidence for their effectiveness

- identify research gaps.

## Method

We used the Arksey and O'Malley framework and Levac's advanced methodology, to conduct the scoping review [6, 7].

The Jokanovic review [4] was used as a source for systematic reviews up to and including 2015. A supplementary search of Embase and MEDLINE databases using the OVID platform identified reviews published between January 2016 and January 2023 (the time of the search) (S1 File). An (updated) measurement tool to assess systematic reviews (AMSTAR 2) critical appraisal tool was used to assess the quality of the included systematic reviews [8]. The

systematic reviews were critically appraised by only one reviewer (MC), as this is not essential for a scoping reviewer. Levac et al recommend quality assessment to increase the confidence in conclusions made about gaps in the literature [7].

The inclusion and exclusion criteria for the systematic reviews is as follows:

- All adult ($\geq$18 years) recipients of medication reviews regardless of the setting or medical history.

- Participants received a medication review. To be included, at least 50% of the primary studies must have utilised a medication review as the intervention.

- Results and/or discussion make specific reference to the implementation and impact of medication review.

- Reviews reported all outcomes.

- Systematic reviews containing all types of studies.

  Reviews were excluded if:

- full text was not available.

- pharmacists did not have a leading role in delivering the interventions.

- not available in the English language; time and financial constraints did not allow for translation from other languages.

Data extracted by MC included the number and design of included studies, population, setting, main results, and description of intervention. This was done using a bespoke data collection form, which was tested using papers included in the Jokanovic review [4]. The systematic reviews were studied for details of the nature of the intervention and whether the authors of the reviews reported on these components during their results, discussion, or conclusion. The components of interest of the medication review were type of review (PCNE level 2 or 3 [9]), mode of delivery (e.g. face-to-face, telephone), setting (e.g. community pharmacy, hospital), duration (how long), intensity (how often), and collaboration with other healthcare professionals. A PCNE level 2 reviews medication history available in the pharmacy alongside information from clinical records or obtained directly from the patient. A level 3 review utilises information obtained from medication history, clinical records and directly from the patient. The main results at the systematic review level were summarised and reported as evidence of no effect, uncertain effect, or evidence of effect.

To address the final objective, the authors' commentary on gaps in the literature and recommendations for future research were extracted and summarised.

## Results

The searches yielded 85 titles after deduplication. Following the application of inclusion and exclusion criteria, 24 systematic reviews were included. Fig 1 outlines the selection process for this scoping review.

### Description of the extent and range of pharmacist-led medication reviews

Twelve systematic reviews included only randomised controlled trials (RCTs) [11–22], one included only non-randomised studies [23] and all other reviews included both randomised and non-randomised trials [24–34].

Original studies included in the 24 systematic reviews were conducted in a wide range of countries across five continents (S1 Table). Eight reviews included populations restricted by

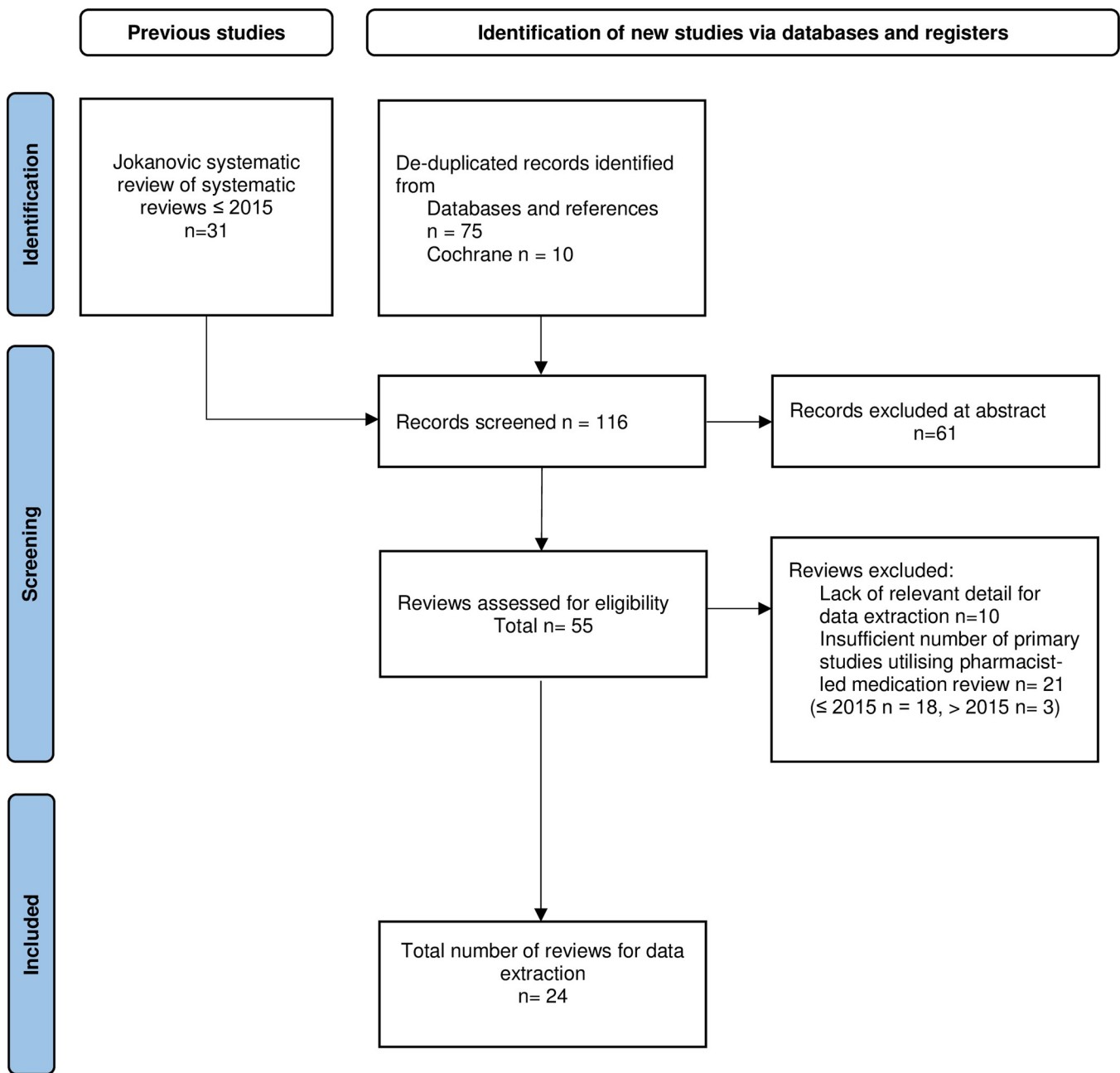

**Fig 1. Flow diagram for literature review [10].**

age: seven included studies with older adults [11, 12, 14, 16, 22, 24, 27] and one children and adolescents [25]. Two reviews included patient populations with cardiovascular disease and/ or other chronic conditions [18, 19]. Four reviews included restricted settings: two included studies based only in hospital [21, 34], one in care homes [16], and another in ambulatory care [33].

The full AMSTAR 2 assessment is reported in S2 Table. Seventeen of the 24 reviews were rated as critically low confidence in the methods and results, two as low, three as moderate, and two as high. Seventeen reviews did not report a registered protocol and fourteen did not

provide sufficient justification of the exclusion of studies; these are critical domains. Where one critical flaw is observed, these reviews can only be low confidence. Where more than one critical flaw was observed, these reviews were assessed as critically low. If authors had indicated a published protocol and included more detailed description of excluded studies, overall confidence in results would have been greater.

## Reported nature of the medication review

The components of the nature of the medication review we were interested in were type of medication review, intensity, duration, mode of delivery, setting, and collaboration. The extent to which these were discussed varied greatly, with all except one review [26] discussing at least one component, with collaboration between healthcare professionals reported most often. Most reviews did not explore intervention components to any great degree. Two reviews aimed to investigate the effectiveness of collaboration between pharmacists and doctors and the effect on outcomes [14, 29], one sought to examine the components of medication reviews to better support a specific population [23], whilst another described the presence of specific activities reported in the medication reviews [19]. A further three reviews reported on one or more of intensity, type of MR, and mode of delivery in their results [13, 20, 30]. The authors of the remaining systematic reviews referred to the nature of the intervention in the discussion only.

A meta-analysis by Martinez-Mardones et al. [19] examined the effects of different components of MRs such as access to clinical records, education, self-monitoring, lifestyle advice, and medicines-related problems. It showed that MRs that includes a patient interview, in addition to access to medication and clinical data, led to greater reductions in blood pressure, glycated haemoglobin, and cholesterol, than an MR which did not include a patient interview or access to clinical records. Hatah et al. undertook a subgroup analysis and reported that face-to-face mediation reviews with or without access to full clinical notes reduced unplanned hospital admissions more than reviews merely assessing issues relating to patients' medication-taking behaviour [30]. Bulow et al. were unable to determine the effect of MR components on reported outcomes [21].

Hikaka et al.'s description of included studies identified components of the nature of the intervention such as type of review (medicines use review or comprehensive MR), setting (home or pharmacy), and delivery mode (face-to-face or telephone) [23]. The inclusion of low-quality and observational studies, in addition to the varying outcomes reported, made it difficult to establish the effect of the different components on outcomes [23].

Geurts et al. and Kwint et al. aimed to examine whether collaboration between pharmacists and general practitioners (GPs) influenced patient outcomes [14, 29]. These reviews observed that GP implementation of pharmacist recommendations was more likely with increased collaborative working between pharmacists and general practitioners. Tan et al. highlighted that a positive effect on patient outcomes was more likely to be observed if the MR was combined with interprofessional face-to-face communication [15]. Jokanovic concluded that MRs conducted by medical practice-based pharmacists were associated with higher rates of implementation of recommendations [32].

Huiskes' review included MRs delivered during a short intervention period ($\leq$3 months) [17]. They recommended the development and evaluation of interventions with multiple contacts between practitioner and patient. Rollason also suggested that more than one MR contact could lead to better outcomes [24].

Bulow reports that MRs in combination with other interventions, e.g., patient education and medication reconciliation, reduced hospital readmissions when compared to usual care, but standard MRs did not [21].

## Reported outcomes and evidence for effectiveness of medication reviews

The authors' outcomes of interest are reported in S1 Table. The most frequently reported outcome was the effects of medication reviews on healthcare utilisation, which includes hospital admissions, re-admissions, and access to primary care physicians. Eight of the seventeen reviews [11, 14, 15, 22, 23, 29, 30, 34] reported that MRs no evidence of a positive effect on healthcare utilisation. Seven reviews [12, 16, 20, 25, 31–33] reported mixed evidence of a positive effect on healthcare utilisation and that the effect of MRs was uncertain, whilst one high-quality and one critically low review [21, 26] reported a reduction in hospital (re) admissions. Four reviews reported mixed evidence of MR effect on medicines costs [16, 30–32].

Effects of MRs on medication adherence was the second most reported outcome. Two critically low reviews reported that MRs improved adherence [23, 27], another stated that there was no evidence of a positive effect [14], whilst the remaining reviews were uncertain of the impact due to mixed results in primary studies [11, 17, 20, 22, 25, 30, 32, 33].

Fourteen reviews reported at least one patient-orientated outcome [11, 13, 14, 16, 17, 21–23, 25, 29–33], the most frequent being quality of life (12/14). Three reviews [13, 14, 31] reported that the MR did not improve quality of life, but the remaining nine reviews showed mixed results. Eight reviews reported mixed outcomes for patient satisfaction following MRs [11, 13, 15, 22, 23, 25, 31, 33]. Evidence for MRs having a positive effect on medication-related problems is inconclusive due to mixed results. Reviews reported that pharmacists were able to identify medicines-related problems [11, 12, 14, 16, 17, 22, 23, 25, 34]; this was quantified in five of these reviews [11, 14, 16, 23, 29] with the remaining reviews reporting this in results or discussion. However, the resolution of medicines-related problems was not reported or yielded mixed results.

The systematic reviews reported mixed effects on clinical outcomes. Four reviews reported reductions in blood pressure and cholesterol levels; Al-babtain, Martinez-Mardonez, and Tan reported improvements in diabetes biomarkers [15, 19, 20, 30]. Huiskes et al. reported that MRs had a positive effect on falls reduction [17]. Other reviews reported mixed results on the effect of MR on clinical outcomes [12, 13, 18, 29, 31, 33]. The cost-effectiveness of MRs was reported in five systematic reviews [11, 18, 29, 30, 32] but was identified as an area for future research in another four [25–27, 29].

## Gaps in the literature as identified by authors of the systematic reviews

All systematic reviews except two [14, 31] reported that MRs had a positive effect on at least one reported outcome. However, evidence for the effectiveness of the clinical effect of MRs is not conclusive. Some authors stress the need to shift focus from current outcomes, such as documenting the number of medication changes, to clinical measures and outcomes that impact patients, such as medication-related morbidity [11, 12, 14, 16, 24, 26, 28]. Tan et al. reported that variations in measured outcomes make it challenging to compare results and suggest standardisation of outcome measures [15].

Several authors commented on the challenge of data analysis given the heterogeneity of the interventions [14–17, 31, 32], and Alldred et al. reported challenges in conducting subgroup analyses for professional and organisation interventions [16]. Hatah et al. concluded that further research is needed to examine the impact of different types of MR on patient outcomes [30]. Holland et al. and Martinez-Mardones et al. highlighted the need for a well-defined medication review, and Kwint recommended the identification of the key components of the MR [11, 14, 19]. Geurts highlighted the need for a system that classifies the activities undertaken in an MR that can be used across countries [29].

## Discussion

### Statement of key findings

This scoping review identified 24 systematic reviews that reported significant variation in evidence for the effectiveness of pharmacist-led MRs. A high-quality review undertaken in care home facilities [16] reported that MRs lead to an improvement and resolution of medicines-related problems such as potential interactions, or inappropriate dose or indication. Another high-quality review [21] concluded that MRs in hospital settings can reduce healthcare utilisation, i.e., hospital (re)admissions, but this reduction was not found in care home settings [16]. Moderate and low-quality reviews undertaken in community pharmacy and/or ambulatory care reported improvements in clinical outcomes, namely, blood pressure, cholesterol, and glycated haemoglobin [19, 20]. Evidence of the effectiveness of MRs on other outcomes across settings and patient populations is uncertain. Most reviews (71%) had a critically low AMSTAR2 rating, meaning low confidence in the results. However, had protocol papers been referenced in five reviews, the number of reviews assessed as high or moderate would have risen to 29% (from 21%), and those critically low reduced to 59%. Many reviews did not report the nature of the intervention; therefore, it is difficult to explain the variation in outcomes observed.

### Strengths and limitations

This scoping review provides an overview of the systematic reviews that have been conducted researching pharmacist medication reviews in all settings. The search terms used in this review were limited; additional terminology for MRs, for example, drug review, were not used. This may have limited the number of search findings and may have introduced selection bias whilst identifying papers. The Jokanovic systematic review of systematic reviews, which was used as a source for reviews published before 2015, included only MRs undertaken in community settings and excluded care home settings [4]. Consequently, we may not have included any reviews of MRs in care home settings published before 2015. Paper screening and data extraction was only undertaken by one researcher, therefore there is a greater margin of error with screening and extracting data compared to these activities being performed by multiple researchers.

### Interpretation

This scoping review has identified uncertain evidence for the effectiveness of pharmacist-led medication reviews across different patient populations. There was significant variability in the reporting of outcomes associated with pharmacist-led medication reviews; the two most reported outcomes were healthcare utilisation and adherence. However, even these were reported using a variety of methods, making definitive conclusions difficult to draw. Researchers should consider the appropriateness of the outcomes and measures used to assess the effectiveness of MRs. As greater work is undertaken by researchers to understand the range of outcomes associated with a medication review (undertaken by any professional) [35, 36], it may become easier to understand the impact of this type of intervention across multiple studies. Many systematic reviews included evaluation of other interventions alongside medication reviews, and the outcomes and conclusions may have been influenced by the co-interventions.

Alongside a focus on outcomes, understanding the effectiveness of medication reviews requires a clear understanding of the intervention itself and its implementation. Definitions and detailed descriptions of medication reviews were absent in most reviews. The National Institute of Health and Care Excellence (NICE) and the Pharmaceutical Care Network Europe

(PCNE) have published definitions of structured medication reviews [37, 38], but these were rarely cited in the reviews. As some systematic reviews were published before publication of these definitions, this may account for their lack of citation. This lack of definition and therefore description of the MR, may in turn contribute to the mixed results of reported outcomes. The lack of reported MR definitions supports the conclusions of an overview review by Silva et al. that the substantial heterogeneity in definitions, terminologies, and approaches to the delivery of medication reviews impacts the ability to assess the strength of evidence for the effectiveness of MRs [39].

Although intervention descriptions were poorly reported, components that appear to positively influence outcomes include face-to-face contact with patients, pharmacist access to clinical notes, and collaborative working with physicians [14, 19, 29, 30]. The lack of confidence in the results of these studies means that future randomised or non-randomised studies should explore this conclusion. Authors reported that variation in the approaches to medication reviews impacts the ability to evaluate their effectiveness [14–17, 31, 32], therefore, standardisation of the MR would be beneficial for future evaluation. Geurts [29] recommended a classification system for medication reviews and Alharti et al. aimed to identify MR activity terms and definitions reported in primary studies [40]. They concluded that developing an international taxonomy for medication reviews and their activities would be beneficial rather than creating a standardised intervention for use in all settings.

## Further research

Variations in the approach to and description and attributes of medication reviews may lead to inconsistent results. Therefore, exploring individual components of medication reviews and linking these to outcomes may be a better approach to appraise their effectiveness than reporting results alone. Silva et al. concluded that an international agreement on the medication review process was necessary and Alharti et al. advanced this conclusion by identifying terms used to describe MR activity [39, 40]. Bulow et al concluded that it is uncertain what type of medication review is the most effective [21]. This scoping review will inform another stage of research which may have implications for policy and practice. Future systematic reviews could reduce the challenges of interpretation of the results due heterogeneity of the intervention by having more restrictive inclusion criteria, for example, limiting the included studies to specific outcomes and validated measures, such as health-related quality of life, medicines appropriateness, or to those where there a good description of the intervention. There is also an opportunity for a future systematic review to focus on the different components/attributes of medication reviews and determine which of these lead to particular outcomes. A rapid review of evidence of clinical pharmacy services in the UK reported that implementation of medication use reviews was unstandardised, which led to a disparity in delivery [41]. This review concluded that the evidence for the cost-effectiveness of the review was lacking. This rapid review reinforces the conclusion of this scoping review, that there is a need to standardise reporting of interventions in order that coherent conclusions can be drawn about the effectiveness of medication reviews. This will then aid policy makers and practitioners in implementing medication reviews which are more likely to yield better outcomes for patients.

## Conclusions

We included twenty-four systematic reviews which showed that the evidence of effectiveness of medication reviews across settings and patient populations is uncertain. Quality assessment of the reviews rated the majority as low or critically low confidence in the results; therefore, these should be interpreted with caution. Reporting of the nature of medication reviews lacked

clarity, therefore it is difficult to ascertain what is happening during the intervention. This lack of clarity in turn makes it difficult to explain the inconsistent outcomes observed. As medication reviews are widely implemented in practice, it would be useful to explore further the nature of the intervention and link their components to outcomes. As researchers gain a better understanding of these components and hypothesize what works for whom, when, and how, this can inform future implementation of medication reviews.

## Supporting information

**S1 File. Search strategy.**
(DOCX)

**S1 Table. Study characteristics and results.**
(DOCX)

**S2 Table. Full AMSTAR rating of included studies.**
(DOCX)

## Author Contributions

**Data curation:** Miriam Craske.

**Formal analysis:** Miriam Craske.

**Supervision:** Wendy Hardeman, Nicholas Steel, Michael James Twigg.

**Writing – original draft:** Miriam Craske.

**Writing – review & editing:** Miriam Craske, Wendy Hardeman, Nicholas Steel, Michael James Twigg.

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
