## [Decision Letter · Decision Letter 0]

23 Nov 2023

PONE-D-23-33977Pharmacist-led medication reviews: A scoping review of systematic reviewsPLOS ONE

Dear Dr. Craske,

Thank you for submitting your manuscript to PLOS ONE. After careful consideration, we feel that it has merit but does not fully meet PLOS ONE’s publication criteria as it currently stands. Therefore, we invite you to submit a revised version of the manuscript that addresses the points raised during the review process.

We look forward to receiving your revised manuscript.

Kind regards,

Nasser Hadal Alotaibi

Academic Editor

PLOS ONE

Journal Requirements:

3. Please ensure that you include a title page within your main document. You should list all authors and all affiliations as per our author instructions and clearly indicate the corresponding author.

4. Please include your tables as part of your main manuscript and remove the individual files. Please note that supplementary tables (should remain/ be uploaded) as separate "supporting information" files

Reviewers' comments:

Reviewer's Responses to Questions

**Comments to the Author**

1. Is the manuscript technically sound, and do the data support the conclusions?

Reviewer #1: Partly

Reviewer #2: Yes

2. Has the statistical analysis been performed appropriately and rigorously? 

Reviewer #1: Yes

Reviewer #2: N/A

3. Have the authors made all data underlying the findings in their manuscript fully available?

Reviewer #1: Yes

Reviewer #2: Yes

4. Is the manuscript presented in an intelligible fashion and written in standard English?

Reviewer #1: Yes

Reviewer #2: No

5. Review Comments to the Author

Reviewer #1: The manuscript titled as Pharmacist-led medication reviews: A Scoping Review of systematic reviews submitted to PLOS ONE with manuscript number PONE-D-23-33977 for possible publication is not suitable for the publication as such due to following points. Accepted after the following MINOR revisions.

General Comments:

The manuscript provides a comprehensive overview of pharmacist-led medication reviews. The structure and content are well-organized, covering the most important aspects of the literature. However, the following critical points require further attention to enhance the validity and impact of the findings.

• The Scoping review method was outlined but the manuscript would benefit from more explicit details regarding the search strategy, inclusion and exclusion criteria, selection process, and data extraction process. This would significantly bolster the credibility and transparency of this manuscript’s findings.

• It is important to clearly define and explain the components of the medication review interventions investigated in the systematic reviews. The absence of this information limits the understanding of the elements contributing to outcomes.

• A significant part of the systematic reviews received low AMSTAR 2 ratings, which raises concerns about their methodological quality. This directly impacts the reliability and trustworthiness of the conclusions drawn from these reviews. This can be addressed by discussing the potential reasons for such poor ratings or by suggesting strategies for improving the quality of the reviews.

• The manuscript highlights gaps and challenges in the reviews (e.g., interpretation of results due to heterogeneity among interventions and outcomes) but doesn’t suggest a strategy to address them. Suggestions for future research directions and methodologies to overcome such issues can enhance the impact of the manuscript.

• The manuscript also identifies the lack of consistent reporting and standardization across reviews as a major limitation but does not provide detailed suggestions on how this standardization can be achieved. Citing the NICE and PCNE definitions is a good start, but more emphasis on the necessity for establishing standardized reporting is required.

• In the conclusion the uncertainty of the evidence is highlighted, however, it would be beneficial to provide clear implications for practice and policy. i.e. How can these findings guide the implementation of pharmacist-led medication review in real-life scenarios?

What are the practical implications for healthcare professionals, policymakers, and patients?

The manuscript effectively reviews the existing literature on the topic. Incorporation of the above-mentioned suggestions will strengthen the manuscript and refine the review’s contributions to the field and future research and practice.

Reviewer #2: Thank you for addressing this important topic. Here are some specific comments and suggestions for improvement:

Abstract:

• In the background section, consider providing more context for the importance of "medication" reviews (note the typo in "edication").

• In the aim section, it would be beneficial to be more specific by stating that the review aims to examine the "extent, range, and quality" of systematic reviews on pharmacist-led medication reviews.

• In the methodology section, consider upgrading by providing detailed information on the databases searched, inclusion/exclusion criteria, and the criteria used to assess the quality of the systematic reviews.

• In the conclusion section, it would be more impactful to highlight the main takeaways from the review and emphasize the need for further research in this area.

Introduction:

• The introduction is clear and effectively sets the stage for the importance of medication reviews.

• Consider incorporating a more explicit and succinct statement of the research question or objective to enhance clarity.

Methodology:

• Positive feedback is provided for the methodology, indicating satisfaction with the methods used.

Overall Impression:

• The paper is well-researched and provides a thorough examination of existing systematic reviews on pharmacist-led medication reviews.

• While the language is generally clear, attention to detail in sentence construction and grammar is recommended.

Recommendations:

• Implement the suggested improvements in the abstract, ensuring clarity on the aim, methodology, and conclusion.

• In the introduction, incorporate a more explicit statement of the research question or objective.

• Pay close attention to detail in sentence construction and grammar throughout the paper.

Thank you for your efforts, and I look forward to seeing the revisions.

6. PLOS authors have the option to publish the peer review history of their article (what does this mean?). If published, this will include your full peer review and any attached files.

Reviewer #1: No

Reviewer #2: **Yes: **Sajidur Rahman Akash

---

## [Author Response · Author response to Decision Letter 0]

6 May 2024

In response to the journal requirements highlighted,

1. The manuscript has been updated to reflect PLOS ONE’S style requirements

2. Funding information has been removed from the manuscript

3. The title page has been included within in the main document

4. Figures have been added to the manuscript with supplementary files as “supporting information”

5. Captions for the supporting information have been updated to reflect style requirements.

6. The reference list has been reviewed. Reference [1], Network Contract Direct Enhanced Service: Contract specification 2019/20 has been replaced with the current updated Network Contract Directed Enhanced Service Contract specification 2023/24-PCN Requirements and Entitlements. Reviewer One

The manuscript provides a comprehensive overview of pharmacist-led medication reviews. The structure and content are well-organized, covering the most important aspects of the literature. 

Thank you

However, the following critical points require further attention to enhance the validity and impact of the findings.

• The Scoping review method was outlined but the manuscript would benefit from more explicit details regarding the search strategy, inclusion and exclusion criteria, selection process, and data extraction process. This would significantly bolster the credibility and transparency of this manuscript’s findings.

The research strategy for the scoping review is described in S1 file. However, we recognise that the inclusion and exclusion criteria were difficult to find. Therefore, we have made this clearer. (See lines 98-111)

 The inclusion and exclusion criteria for the systematic reviews is as follows: 

• All adult (≥18 years) recipients of medication reviews regardless of the setting or medical history.

• Participants received a medication review. To be included, at least 50% of the primary studies must have utilised a medication review as the intervention. 

• Results and/or discussion make specific reference to the implementation and impact of medication review. 

• Reviews reported all outcomes.

• Systematic reviews containing all types of studies.

Reviews were excluded if:

• full text was not available.

• pharmacists did not have a leading role in delivering the interventions.

• not available in the English language; time and financial constraints did not allow for translation from other languages.

• It is important to clearly define and explain the components of the medication review interventions investigated in the systematic reviews. The absence of this information limits the understanding of the elements contributing to outcomes.

Thank you for bringing this to our attention. Some additional information has been added to lines 117-124 for clarity 

“The components of interest of the medication review were type of review (PCNE level 2 or 3[9]), mode of delivery (e.g. face-to-face, telephone), setting (e.g. community pharmacy, hospital), duration (how long), intensity (how often), and collaboration with other healthcare professionals. A PCNE level 2 reviews medication history available in the pharmacy alongside information from clinical records or obtained directly from the patient. A level 3 review utilises information obtained from medication history, clinical records and directly from the patient. The main results at the systematic review level were summarised and reported as evidence of no effect, uncertain effect, or evidence of effect.”

• A significant part of the systematic reviews received low AMSTAR 2 ratings, which raises concerns about their methodological quality. This directly impacts the reliability and trustworthiness of the conclusions drawn from these reviews. This can be addressed by discussing the potential reasons for such poor ratings or by suggesting strategies for improving the quality of the reviews.

Agree. The quality of the reported studies does impact on the reliability of the conclusions. Potential reasons for poor ratings have been added from line 149 “Seventeen of the 24 reviews were rated as critically low confidence in the methods and results, two as low, three as moderate, and two as high. Seventeen reviews did not report a registered protocol and fourteen did not provide sufficient justification of the exclusion of studies; these are critical domains. Where one critical flaw is observed, these reviews can only be low confidence. Where more than one critical flaw was observed, these reviews were assessed as critically low. If authors had indicated a published protocol and included more detailed description of excluded studies, overall confidence in results would have been greater. The full AMSTAR 2 assessment is reported in S3 table.”

• The manuscript highlights gaps and challenges in the reviews (e.g., interpretation of results due to heterogeneity among interventions and outcomes) but doesn’t suggest a strategy to address them. Suggestions for future research directions and methodologies to overcome such issues can enhance the impact of the manuscript.

Consideration of future research direction has been added, lines 331-343

“Future systematic reviews could reduce the challenges of interpretation of the results due heterogeneity of the intervention by having more restrictive inclusion criteria, for example, limiting the included studies to specific outcomes and validated measures, such as health-related quality of life, medicines appropriateness, or to those where there a good description of the intervention. There is also an opportunity for a future systematic review to focus on the different components/attributes of medication reviews and determine which of these lead to particular outcomes. 

• The manuscript also identifies the lack of consistent reporting and standardization across reviews as a major limitation but does not provide detailed suggestions on how this standardization can be achieved. Citing the NICE and PCNE definitions is a good start, but more emphasis on the necessity for establishing standardized reporting is required.

Additional information has been added, lines 337-343 “ A rapid review of evidence of clinical pharmacy services in the UK reported that implementation of medication use reviews was unstandardised, which led to a disparity in delivery [41]. This review concluded that the evidence for the cost-effectiveness of the review was lacking. This rapid review reinforces the conclusion of this scoping review, that there is a need to standardise reporting of interventions in order that coherent conclusions can be drawn about the effectiveness of medication reviews. This will then aid policy makers and practitioners in implementing medication reviews which are more likely to yield better outcomes for patients.” 

• In the conclusion the uncertainty of the evidence is highlighted, however, it would be beneficial to provide clear implications for practice and policy. i.e. How can these findings guide the implementation of pharmacist-led medication review in real-life scenarios?

What are the practical implications for healthcare professionals, policymakers, and patients?

Thank you for identifying this. This has been addressed in the response to the previous comment.

The manuscript effectively reviews the existing literature on the topic. Incorporation of the above-mentioned suggestions will strengthen the manuscript and refine the review’s contributions to the field and future research and practice.

Reviewer Two

Abstract:

• In the background section, consider providing more context for the importance of "medication" reviews (note the typo in "edication").

• In the aim section, it would be beneficial to be more specific by stating that the review aims to examine the "extent, range, and quality" of systematic reviews on pharmacist-led medication reviews.

• In the methodology section, consider upgrading by providing detailed information on the databases searched, inclusion/exclusion criteria, and the criteria used to assess the quality of the systematic reviews.

• In the conclusion section, it would be more impactful to highlight the main takeaways from the review and emphasize the need for further research in this area.

Thank you for these comments. We have amended the abstract to address these comments.

Line 26 There are different types of medication reviews undertaken in various settings. 

Line 31 “This scoping review aims to explore the extent and range of systematic reviews …”

Lines 35,36 “Systematic reviews were included irrespective of participants, settings or outcomes and were excluded if pharmacists did not lead the delivery of the included interventions.”

Lines 49-53 There is mixed evidence of effectiveness for medication reviews across settings and patient populations. There is limited data about the implementation of medication reviews, therefore is difficult to ascertain which components of the intervention lead to improved outcomes. As medication reviews are widely implemented in practice, further research should explore the nature of the interventions, linking the components of these to outcomes.

Introduction:

• The introduction is clear and effectively sets the stage for the importance of medication reviews.

Thank you

• Consider incorporating a more explicit and succinct statement of the research question or objective to enhance clarity.

A more explicit research question has been added (lines 75-77)

“Therefore, our research question is, what is the systematic review evidence about the nature and effectiveness of medication review interventions conducted by pharmacists, and what are the gaps in research knowledge?”

Methodology:

• Positive feedback is provided for the methodology, indicating satisfaction with the methods used.

Thank you

Overall Impression:

• The paper is well-researched and provides a thorough examination of existing systematic reviews on pharmacist-led medication reviews.

Thank you

• While the language is generally clear, attention to detail in sentence construction and grammar is recommended.

Thank you for bringing these errors to our attention. This has been reviewed and actioned.

---

## [Decision Letter · Decision Letter 1]

19 Aug 2024

Pharmacist-led medication reviews: A scoping review of systematic reviews

PONE-D-23-33977R1

Dear Dr. Craske,

We’re pleased to inform you that your manuscript has been judged scientifically suitable for publication and will be formally accepted for publication once it meets all outstanding technical requirements.

Kind regards,

Nasser Hadal Alotaibi

Academic Editor

PLOS ONE

Additional Editor Comments (optional):

Reviewers' comments:

Reviewer's Responses to Questions

**Comments to the Author**

1. If the authors have adequately addressed your comments raised in a previous round of review and you feel that this manuscript is now acceptable for publication, you may indicate that here to bypass the “Comments to the Author” section, enter your conflict of interest statement in the “Confidential to Editor” section, and submit your "Accept" recommendation.

Reviewer #1: All comments have been addressed

2. Is the manuscript technically sound, and do the data support the conclusions?

Reviewer #1: Yes

3. Has the statistical analysis been performed appropriately and rigorously? 

Reviewer #1: Yes

4. Have the authors made all data underlying the findings in their manuscript fully available?

Reviewer #1: Yes

5. Is the manuscript presented in an intelligible fashion and written in standard English?

Reviewer #1: Yes

6. Review Comments to the Author

Reviewer #1: The manuscript titled as Pharmacist-led medication reviews: A Scoping Review of systematic reviews submitted to PLOS ONE with manuscript number PONE-D-23-33977 for possible publication is now suitable for the publication as all the points previously raised has been well addressed. The manuscript is updated now and well modified as suggested. Now, its recommended for the Acceptance of this manuscript.

7. PLOS authors have the option to publish the peer review history of their article (what does this mean?). If published, this will include your full peer review and any attached files.

Reviewer #1: No

---

## [Editor Report · Acceptance letter]

27 Aug 2024

PONE-D-23-33977R1 

PLOS ONE

Dear Dr. Craske, 

I'm pleased to inform you that your manuscript has been deemed suitable for publication in PLOS ONE. Congratulations! Your manuscript is now being handed over to our production team.

Kind regards, 

on behalf of

Dr. Nasser Hadal Alotaibi 

Academic Editor

PLOS ONE